# pH-Responsive Succinoglycan-Carboxymethyl Cellulose Hydrogels with Highly Improved Mechanical Strength for Controlled Drug Delivery Systems

**DOI:** 10.3390/polym13183197

**Published:** 2021-09-21

**Authors:** Younghyun Shin, Dajung Kim, Yiluo Hu, Yohan Kim, In Ki Hong, Moo Sung Kim, Seunho Jung

**Affiliations:** 1Center for Biotechnology Research in UBITA (CBRU), Department of Bioscience and Biotechnology, Konkuk University, Seoul 05029, Korea; syh4969@naver.com (Y.S.); dajung903@naver.com (D.K.); lannyhu0806@hotmail.com (Y.H.); shsks1@daum.net (Y.K.); 2Covergence Technology Laboratory, Kolmar Korea, 61, Heolleung-ro-8-gil, Seocho-gu, Seoul 06800, Korea; inkiaaa@kolmar.co.kr; 3Macrocare, 32 Gangni 1-gil, Cheongju 28126, Korea; rnd@macrocare.net; 4Center for Biotechnology Research in UBITA (CBRU), Department of Systems Biotechnology & Institute for Ubiquitous Information Technology and Applications (UBITA), Konkuk University, Seoul 05029, Korea

**Keywords:** hydrogels, carboxymethyl cellulose, succinoglycan, metal coordination, drug delivery, swelling properties

## Abstract

Carboxymethyl cellulose (CMC)-based hydrogels are generally superabsorbent and biocompatible, but their low mechanical strength limits their application. To overcome these drawbacks, we used bacterial succinoglycan (SG), a biocompatible natural polysaccharide, as a double crosslinking strategy to produce novel interpenetrating polymer network (IPN) hydrogels in a non-bead form. These new SG/CMC-based IPN hydrogels significantly increased the mechanical strength while maintaining the characteristic superabsorbent property of CMC-based hydrogels. The SG/CMC gels exhibited an 8.5-fold improvement in compressive stress and up to a 6.5-fold higher storage modulus (G′) at the same strain compared to the CMC alone gels. Furthermore, SG/CMC gels not only showed pH-controlled drug release for 5-fluorouracil but also did not show any cytotoxicity to HEK-293 cells. This suggests that SG/CMC hydrogels could be used as future biomedical biomaterials for drug delivery.

## 1. Introduction

Hydrogels are a three-dimensional hydrophilic polymer capable of absorbing large amounts of water [1]. Due to their unique properties, hydrogels are widely used in pharmaceutical, tissue engineering, biomedical, cosmetic, and drug delivery systems [2,3,4,5,6]. Natural polymers and synthetic polymers are used to manufacture hydrogels. Compared to synthetic-based polymers, natural polymers tend to be environmentally friendly, reproducible, and biocompatible [7]. Among them, natural polysaccharides derived from microorganisms are very ideal candidates for the fabrication of new hydrogels because they have biodegradability, versatility, and biocompatibility [8]. In many studies, bacterial polysaccharides such as chitosan [9], starch [10], xanthan gum [11], gellan gum [12], and alginate [7] have been reported as hydrogel components. Particularly, acidic bacterial polysaccharide-based hydrogels have a very wide application field because they can successfully perform drug delivery depending on the pH [13].

Succinoglycan is an acidic exopolysaccharide (EPS) derived from soil microorganisms *Sinorhizobium* and *Agrobacterium* [14,15]. It plays an important role in the development of the root nodule between bacteria and the Alfalfa legume [14]. Succinoglycan is a polysaccharide with repeating octasaccharides composed of seven glucose residues and one galactose residue. They have succinate, pyruvate, and acetate groups as non-carbohydrate substituents [16]. Due to the functional groups with a carboxyl group, it is easy to crosslink with metal cations such as Fe^3+^ and Cr^3+^ [17], and is very sensitive to pH [18]. In addition, succinoglycan can maintain the physical consistency of its physical properties even under extreme conditions such as high temperature, high shear rate, high salinity, or ionic concentration [19,20]. In particular, the high heat stability of succinoglycan showed mass stability of about 60%, even at 600 °C, as measured using thermogravimetric analysis (TGA) [17].

Carboxmethyl cellulose (CMC), a derivative of cellulose, is synthesized by the reaction between cellulose and chloroacetic acid [21]. Unlike cellulose, CMC is relatively soluble in water and can absorb large amounts of water. Due to these characteristics, CMC has numerous potentials as a superabsorbent hydrogel, such as controlled fertilizer and agrochemicals, drug delivery, wound dressing, and tissue engineering [22,23,24]. However, despite these advantages, CMC hydrogels are limited in their application due to the critical disadvantage of weak mechanical strength. Therefore, various multi-component CMC hydrogel systems have been reported for the purpose of improving mechanical strength [25]. For example, the addition of polyacrylamide, acrylic acid, and carboxymethyl β-cyclodextrin increased the mechanical strength of CMC-based hydrogels [26,27,28]. However, these synthetic-based polymers have disadvantages of low biodegradability, versatility, and biocompatibility compared to natural polysaccharides derived from microorganisms [8]. In addition, since they are not acidic polysaccharides, there is a limit to pH-dependent drug delivery. Therefore, CMC-based hydrogels using acidic polysaccharides have been reported, but there is a disadvantage of being in the form of beads or modified forms [29,30,31,32].

To our knowledge, there have been no reports of non-beaded CMC-based hydrogels capable of pH-responsive drug delivery using succinoglycan, an unmodified bacterial polysaccharide to significantly improve the mechanical strength of hydrogels while maintaining characteristic superabsorbency. We hypothesized that bacterial succinoglycan could increase the mechanical strength of CMC-based hydrogels because succinoglycan can maintain a stable consistency of physical properties even in extreme environments [7,17].

Here, succinoglycan (SG), an unmodified natural polysaccharide, was successfully used to increase the mechanical strength of the CMC hydrogel. Since polysaccharide-based IPN (interpenetrating polymer network) hydrogels provide excellent biocompatibility, mechanical strength, and excellent phase stability [33,34], an SG/CMC IPN hydrogel (SG/CMC gel) was fabricated by cross-linking the hydroxyl groups and carboxyl groups present in both SG and CMC through Fe^3+^ ions. These structures were characterized using Fourier Transform Infrared (FTIR) Spectroscopy, Thermogravimetric Analysis (TGA), Field Emission Scanning Electron Microscopy (FE-SEM), a rheology test, and a compressive test. We also investigated the pH-responsive drug release properties using 5-fluorouracil as a model drug.

## 2. Materials and Methods

### 2.1. Materials

The bacterial strain (*Sinorhizobium meliloti* Rm1021) was supplied by the Microbial Carbohydrate Resource Bank (MCRB) at Konkuk University (Seoul, Korea). CMC (Mw = 250,000 g/mol with degree of substitution 0.7–based on manufacturer’s data) was obtained from Sigma Aldrich (St. Louis, MO, USA). Iron (III) chloride hexahydrate (97.5%) was purchased from Daejung Chemicals & Metals Co., Ltd. (Siheung-si, Korea) and 5-fluorouracil (5-FU) purchased from Sigma Aldrich (St. Louis, MO, USA). All other chemicals were of analytical grade and used without further purification.

### 2.2. Growth Conditions and Production and Preparation of Succinoglycan

The isolation and purification of succinoglycan from *S. meliloti* Rm1021 was performed as previously described [18]. Bacteria were cultured in medium comprised of d-mannitol (10 g/L), glutamic acid (1.5 g/L), K_2_HPO_4_ (5 g/L), KH_2_PO_4_ (5 g/L), MgSO_4_·7H_2_O (0.2 g/L), and CaCl_2_·2H_2_O (0.04 g/L), which was adjusted to a pH of 7.00 at 30 °C for 7 days with shaking (180 rpm). After, cells were centrifuged at 8000× *g* for 15 min at 4 °C and the supernatant was collected. To obtain succinoglycan, three volumes of ethanol were added to the supernatant. Furthermore, the precipitated succinoglycan was dissolved in distilled water and dialyzed (MWCO 12–14 kDa, distilled water for 3 days). After collection, succinoglycan purified via dialysis was lyophilized for later use. The molecular weights of succinoglycan were estimated via gel permeation chromatography (GPC) analysis. GPC was performed using a Waters Breeze System equipped with a Waters 1525 Binary pump and a Waters 2414 refractive index detector and was performed at 30 °C with a flow rate of 0.8 mL min^−1^ using 0.02 N sodium nitrate as a solvent. The molecular weight (Mw) of succinoglycan, as estimated via GPC, is 1.8 × 105 Da.

### 2.3. Preparation of Fe^3+^-Crosslinked Layered-SG/CMC IPN Hydrogels

CMC and lyophilized SG were dissolved in distilled water at room temperature and stirred to obtain a clear polymer solution, respectively. As it is difficult to obtain a uniform hydrogel during the manufacturing process by adding Fe^3+^ solution to the polymer solution, we performed with reference to the preparation of a layered hydrogel [35]. To prepare a hydrogel, the prepared SG solution was added to the CMC solution and stirred for 30 min. The resultant polymer mixture solution was poured into a mold. After that, the aqueous polymer mixture solution was poured into special molds with circular grooves (20 cm in diameter, 10 cm in height) and soaked in aqueous iron (III) chloride (FeCl_3_) solution (30 mM) for 18 h. These samples were completely immersed in deionized water and the deionized water was sufficiently changed to remove the nomadic trivalent iron ions from the hydrogel via diffusion of ion. SG/CMC IPN hydrogels obtained from circular groove were labeled as SxCx gels, where x = 1, 2, and 3 represent the relative proportions occupied by the polymer solution, respectively. The composition of SG, CMC, and SG/CMC IPN hydrogels are shown in Table 1.

### 2.4. Characterization of Fe^3+^-Crosslinked Layered SG/CMC IPN Hydrogels

The FTIR spectra of each sample were taken using an FTIR spectrometer (Spectrum Two FTIR, Perkin Elmer), and obtained with a resolution of 0.5 cm^−1^ using 8 scans and a wavenumber range of 4000–600 cm^−1^. All samples were mixed with KBr powder in a certain ratio (1:100) and then compressed to form pellets.

### 2.5. Thermogravimetric Analysis (TGA)

The hydrogels were washed by soaking in an excess of 70% ethanol for 2 days and were then dried at 60 °C and lyophilized. Lyophilized hydrogels were grinded in a blender. Thermogravimetric Analysis (TGA) was performed using a Perkin Elmer Pyris1 thermogravimetric analyzer. The thermal analyzer was operated under nitrogen atmospheric pressure and regulated by compatible PC commands. The dried sample (10 mg) was placed in a crucible and heated in an enclosed system with a linear temperature increase at a rate of 10 °C/min over a temperature range of 25 to 600 °C.

### 2.6. Field Emission Scanning Electron Microscopy (FE-SEM) Analysis

The cross-sectional morphology of SG/CMC IPN hydrogels were observed using FE-SEM (JSM-7800F Prime, JEOL Ltd., Akishima, Japan). The samples were quickly frozen and then lyophilized for 24 h. For observation, the surface of cross-sectioned hydrogel was coated with a thin layer of platinum at 10 mA for 60 s in a vacuum. SEM/EDS analysis of hydrogels was performed with FE-SEM (AURIGA, Carl Zeiss, Oberkochen, Germany) with an Energy selective Backscattered (EsB) detector. All hydrogels were platinum-coated at 10 mA, 120 s before examination using field emission scanning electron microscope.

### 2.7. Compression Test

Compression test was performed by using Instron E3000LT (Instron Inc., Norwood, MA, USA) by preparing a hydrogel disk with a height of 15 mm and a diameter of 20 mm. The sample was placed on a plate and compressed at a rate of 5 mm/min. Compressive stress was recorded when the sample was compressed with a strain of 70%. The measurement was performed in triplicate. All the hydrogels used in the measurements were prepared in the same manner to the rheological analysis.

### 2.8. Rheological Experiments

The rheological properties of the hydrogel were analyzed by oscillating angular frequency sweep and temperature ramp tests using a DHR-2 rheometer (TA Instruments, New Castle, DE, USA) equipped with a 20-millimeter parallel plate. The angular frequency was swept from 0.1 to 100 rad/s at a strain of 0.5% at 25 °C. The hydrogel samples were prepared as disks and were analyzed with a parallel plate measuring 20 mm in diameter, and the gap between plates was adjusted to 1.3 mm. A stress strain amplitude sweep test was conducted on samples from 0.1% to a maximum strain of 100% at 1.0 Hz to determine the limit of the linear viscoelastic region. All hydrogel samples used for the measurement were prepared with the same mold and the same amount of water. Each measurement was performed in triplicate. The temperature ramp test was conducted at a constant angular frequency of 10 rad/s and a constant strain of 1.0%. It was also carried out at 10 to 70 °C with an insulating cover to maintain the temperature.

### 2.9. Equilibrium Swelling Ratio and pH Sensitivity Measurements

The swelling properties of SG/CMC IPN hydrogels were investigated in various pH buffer solutions (pH 2, 4, 6, 8, 10). Each of the dried SG/CMC gels were immersed in pH buffer solutions at 37 °C to induce an equilibrium state. Then, the mass of the swollen hydrogel was weighed at various times after removing excess surface water. The swelling ratio was determined using following equation:(1)$$\mathrm{Swelling}\;\mathrm{ratio}\;\left(\%\right)=\frac{w_{s}-w_{d}}{w_{s}}$$
where w_s_ is the weight of the swollen hydrogel and w_d_ is the weight of the dried hydrogel in a vacuum oven prior to PBS immersion. After separating the hydrogel from the solution for accurate weight measurement, the solution remaining on the surface was gently wiped off using a laboratory tissue. Each measurement was performed in triplicate.

### 2.10. Drug Loading and Drug Release

5-Fluorouracil (5-FU) was used as the model drug to analyze drug release properties of SG/CMC IPN gels in different pH conditions. To put it simply, 1 mg of 5-FU was dissolved in 1 mL of SG/CMC polymer solution. The concentration of 5-FU in the hydrogels was fixed at 1 mg/mL. The obtained hydrogels were incubated in 40 mL of various pH buffer solutions (pH = 1.2, 7.4) at 37 °C with 50 rpm of constant stirring. At each interval time points, 500 µL of released medium was taken out for UV–Vis spectroscopic analysis and then same volume of fresh solution was added. The concentrations of 5-FU were analyzed using a spectrophotometer (UV2450, Shimadzu Corporation, Kyoto, Japan) at a wavelength of 266 nm. The cumulative amount of the drug was calculated by the following equation:(2)$$\mathrm{Cumulative}\;\mathrm{amount}\;\mathrm{of}\;\mathrm{the}\;\mathrm{drug}=C_{n}V+\overset{i=n-1}{\underset{i=1}{\sum }}C_{i}V_{i}$$
where *V* is the release of medium volume, $V_{i}$ is the sampling volume, and $C_{n}$ and $C_{i}$ are the 5-FU concentrations in the release medium and extraction sample. All measurements were conducted in triplicate.

### 2.11. In Vitro Cytotoxicity

The cytotoxicity of the hydrogels was evaluated using WST-8 assays using human embryonic kidney 239 cells (HEK-293, Korean Cell Line Bank, Korea). For direct cytotoxicity tests, HEK-293 cells were seeded into 24-well culture plates at a concentration of 3 × 104 cells per well with minimum essential medium (MEM, WELGENE, Gyeongsan-si, Korea) containing 10% fetal bovine serum and 1% penicillin/streptomycin; then, 5 mg of hydrogel sample was added to the wells and plates were incubated at 37 °C in an atmosphere containing 5% CO_2_. MEM medium was used as a negative control, and 10% (*v*/*v*) dimethylsulfoxide (DMSO) dissolved in the MEM medium was used as a positive control. After a 48-hour incubation, WST-8 assay reagent (QuantiMax, BIOMAX, Seoul, Korea) was added to each well and absorbance was measured at 450 nm. Cell viability was determined using the following formula:(3)$$\mathrm{Cell}\;\mathrm{viability}\;\left(\%\right)=\frac{\mathrm{Absorbance}\;\mathrm{of}\;\mathrm{cells}\;\mathrm{with}\;\mathrm{hydrogel}\;}{\mathrm{Absorbance}\;\mathrm{of}\;\mathrm{negative}\;\mathrm{control}\;\mathrm{cells}}$$

All assays were repeated in triplicate for each sample.

## 3. Result and Discussion

### 3.1. Characterization of SG/CMC IPN Hydrogels

The coordination of the Fe^3+^-crosslinked layered SG/CMC IPN hydrogels were confirmed through FTIR analysis. Figure 1 shows the FTIR spectra of SG, CMC, and S1C1 gels, and Table 2 shows the shift of the characteristic absorption peaks of the FTIR spectra. Here, the coordination of SxCx with Fe^3+^ ions was explained by S1C1 gel, and similar peak shifts occurred in the other SxCx gels. The FTIR spectrum of succinoglycan exhibited absorption peaks at 3326 cm^−1^, corresponding to the –OH stretching bands, and the C=O stretching carbonyl esters of the acetate group showed absorption peaks at 1728 cm^−1^ [36,37,38]. Additionally, the absorption peaks at 1629, 1382, and 1074 cm^−1^ were attributed to the asymmetric C=O stretching vibration of the succinate and pyruvate functional groups, the symmetric stretching vibration of the carboxylate –COO– group in the acid residue, and the asymmetric C-O-C stretching vibration, respectively [36,37]. The FTIR spectrum of CMC showed a broad absorption band at 3408 cm^−1^, which was related to the stretching frequency of the –OH group, and the peaks at 1625, 1422, and 1076 cm^−1^ were related to the asymmetric, symmetric stretching vibrations of the carboxylate groups, and C–C bending of CMC, respectively [39,40].

However, when the Fe^3+^ ions are crosslinked with the polymer, the S1C1 gel absorption peaks shifted to 3433, 1600, and 1049 cm^−1^, respectively. When the polymer coordinated with the Fe^3+^ ions, the –OH stretching band of the polymer mixture significantly red shifted to 3433 cm^−1^. This proves that the –OH group and Fe^3+^ ion of the polymer mixture are coordinated [41]. In addition, the specific peak associated with the carboxyl group was shifted to 1600 cm^−1^, and the asymmetric C-O-C stretching peak by backbone sugar shifted to 1049 cm^−1^. The shift of these peaks is also due to the coordination of the carboxyl group of the polymer mixture with the Fe^3+^ ions [42,43]. In conclusion, the Fe^3+^ ion was strongly coordinated by hydrogen bonds between the hydroxyl groups and carboxyl groups of SG and CMC. A schematic representation of the presumable coordination mechanism of the SG and CMC polymer solutions and FeCl_3_ based on the above findings is shown in Figure 2.

### 3.2. Thermogravimetric Analysis (TGA) and Derivative Thermogravimetry (DTG)

Figure 3 illustrates thermogravimetric analysis (TGA) and the derivative thermogravimetry (DTG) curves of SG gel, S1C1 gel, and CMC gel. As observed, the mass loss seen near the initial 120 °C in all the hydrogels was due to the water evaporation in the hydrogel [26,45]. The SG gel exhibited a 43.4% mass loss in the temperature range of 211–397 °C, and the major DTG picks appeared twice at 264 and 509 °C. The second phase was related with the degradation of the thermally stable structure formed by cross linkage and strong bonds in succinoglycan [17]. Additionally, the weight of the CMC gel decreased significantly between 220 and 495 °C. Thereafter, the maximum DTG peak was observed at 278 °C in the DTG curve, and the degradation steps in this range of CMC gel occurred due to the cellulose backbone cleavage and fragmentation [46]. In comparison, the weight loss in the S1C1 gel was observed at 210 and 392 °C, with maximum DTG peaks at 247 and 523 °C. The weight loss in this range was 38.9%, showing a more moderate weight loss at high temperatures, and it was shown that the formation of a hydrogel with metals reduced the skeletal decomposition of SG and CMC. In addition, a 50% weight loss was observed for SG gel and CMC gel at 386 and 323 °C, respectively, whereas a 50% weight loss was observed for S1C1 gel at 398 °C. Therefore, it was found that the thermal stability of the hydrogel was increased through IPN formation in the process of coordinating SG and CMC with Fe^3+^. This is the first report to increase the thermal stability of a non-beaded CMC-based hydrogel using succinoglycan, an unmodified natural polysaccharide.

### 3.3. Compression Mechanical Properties of SG/CMC IPN Hydrogels

Compression tests were performed to investigate the effect of SG on the mechanical properties of Fe^3+^ crosslinked layered SG/CMC IPN gels. Figure 4 shows the compressive strain curves. Compared to CMC only gels, SG/CMC IPN gels exhibit a higher compressive stress. As the ratio of SG content from S1C3 to S3C1 increased, the compressive stress of SG/CMC IPN gels improved (S3C1 gel: 0.0483 MPa > S1C1 gel: 0.0412 MPa > S1C3 gel: 0.0128 MPa > CMC gel: 0.0057 MPa). This result is exhibited because SG is able to maintain its physical consistency, even under extreme conditions [19,20]. It has been reported that the addition of SG or modified-SG increases the mechanical strength of the hydrogel [17,47]. Therefore, these results indicate that SG effectively enhanced the mechanical strength of the Fe^3+^ crosslinked layered SG/CMC IPN gels. It has been reported to increase the mechanical strength of CMC-based hydrogels using synthetic polymers or modified polymers [26,27,29]. However, it is significant that the mechanical strength of CMC-based hydrogel as a non-bead form was increased by using unmodified natural polysaccharide.

### 3.4. Rheological Behavior of SG/CMC IPN Hydrogels

As shown in Figure 5a, to obtain information on the mechanical properties of the hydrogel, the oscillation angular frequency sweep test was performed. The oscillatory angle frequency sweep experiment was conducted from 0.1 to 100 rad/s at a fixed strain of 0.5. As can be seen in Figure 5, the storage modulus value G′ of all the hydrogels is larger than the loss modulus value G″. This indicates that the hydrogel is not in a fluid sol state, but a stable solid gel state. Additionally, compared with CMC gel, hydrogel containing SG has a larger storage modulus G′ at all frequencies. Comparing the actual G′ value, the CMC gel is about 1010 Pa. However, the G′ value of the S1C1 gel is 6060 Pa, and the G′ value of the S3C1 gel with the highest SG ratio is 6560 Pa. This is an increase of about 6.0 times and 6.5 times, respectively, compared to the CMC gel. This means that the addition of SG to the CMC-based hydrogel improves the mechanical strength, and the mechanical properties can be adjusted according to the amount of SG.

The loss tangent (tan δ) was shown to measure the viscoelastic properties of CMC and SxCx gels (Figure 5b). Figure 5b shows that the SG/CMC IPN gels have low tan δ values at all frequencies. This shows that the gel was stably formed. In particular, the S1C1 gel prepared with a 1:1 ratio of SG and CMC had the lowest tan δ value, indicating the formation of the most mechanically strong hydrogel. Therefore, in subsequent experiments, S1C1 was set as the general SxCx.

Temperature ramp rheological studies of hydrogels were conducted at a constant frequency and strain. In Figure 6, the Tm value where the G″ value is higher than the G′ value did not occur in all the hydrogels. However, it can be seen that the G′ value of the CMC gel became rapidly insignificant. Therefore, the application in various fields is limited. However, in the case of the S1C1 gel with 2% of SG added to the CMC hydrogel, it can be seen that the G′ value is clearly maintained. This means that due to the strong mechanical strength of SG, it did not deteriorate during heating to 70 °C.

### 3.5. Field Emission Scanning Electron Microscopy (FE-SEM)

The FE-SEM image of the SG/CMC gel cross-section is shown in Figure 7. Figure 7a–c were each hydrogel observed at the same magnification and Figure 7d was the S1C1 gel observed at high magnification. A three-dimensional interconnected porous structure was observed on the cross-section surface of the SG gel. These pores can be related to the degree of cross-linking and mechanical strength [47]. The SEM image of the CMC gel shows an increase in the surface area of the hydrogel due to the coarse, small, and frequent porous structure. This large surface area supports the superabsorbency of CMC-based hydrogels by facilitating water molecules to diffuse into the polymer network [48]. On the other hand, the S1C1 gel shows both the interconnected porous structure of the SG gel and the frequent porous structure of the CMC gel. Figure 7d shows the structure of the same S1Cl gel when viewed at a relatively high magnification. As shown in Figure 7d, the layered structure was shown, which may be due to the diffused Fe^3+^ ions cross-linked with the polymer in each layer during SG/CMC IPN gel formation. Fe^3+^ ions form a first cross-link at the interface between FeCl_3_ and SG/CMC, and then permeate from the top to the bottom of the SG/CMC polymer mixture solution to form a layered hydrogel.

EDS confirmed that Fe^3+^ ions were uniformly distributed in the polymer mixture (Appendix A). As shown in Appendix A, when the S1C1 gel is cut into cross-sections, Fe^3+^ ions are evenly distributed from the top to the bottom of the hydrogel. This means that the SG/CMC IPN hydrogel was formed through the uniform penetration of Fe^3+^ ions.

### 3.6. Swelling Behavior of SG/CMC IPN Hydrogels

Figure 8 shows the pH-sensitive swelling behavior of SG/CMC IPN gel at an acidic and alkaline pH. The swelling equilibrium appeared on an average of 6 h, and the hydrogel was immersed in a pH buffer of five points for 2 days for sufficient swelling. SG and CMC-based hydrogels are known to have a large influence on swelling depending on the pH. This pH-dependent swelling behavior is due to the carboxyl groups of SG and CMC. At an acidic pH (pH 2, 4), most carboxylic acid groups are protonated. In addition, the hydrogen bonding interaction between the carboxylate and hydroxyl groups is enhanced. Therefore, the SG/CMC IPN tends to become denser and, consequently, the swelling value decreases [49]. At a higher pH (pH 6, 8), some carboxylic acid groups are ionized; therefore, the hydrogen bonds are broken [49,50]. Therefore, the electrostatic repulsion between the carboxylic acid groups is increased compared to an acidic pH. Therefore, the SG/CMC IPN tends to swell more. The reason that the swelling ratio is reduced in a basic solution (pH 10) is that Fe^3+^ ions prevent the anion–anion repulsion of the carboxylic acid group [50]. Similar results have been reported in many previous reports [49,50,51,52].

Additionally, then, the hydrogels showed a difference in the swelling ratio according to the SG/CMC ratio, but all showed a similar swelling behavior. The three-dimensional structure of the hydrogel prepared using the IPN crosslinking method is expected to provide a wide space for expansion according to the pH because SG and CMC are formed based on hydrogen bonding. These swelling test results suggest that SG/CMC IPN hydrogels using a double-crosslinking strategy can effectively maintain the superabsorbency of CMC while maintaining the swelling changes with pH changes.

### 3.7. Drug Release Profiles of 5-FU

Figure 9 shows the drug release behavior for 16 h in pH 1.2 and pH 7.4 buffer solutions. 5-FU was released within 7 h at pH 1.2, of which 54% was released from the S1C1 gel. In contrast, at pH 7.4, 5-FU was released more rapidly than at pH 1.2. The S1C1 gel completely released the loaded drug 7 h after the start of the release. In Figure 9c, 5-FU was released from the S1C1 gel to clearly show the difference in release according to pH conditions. The buffer was changed from pH 1.2 to pH 7.4 3 h after the start of the release experiment. After changing to the pH 7.4 condition, 5-FU was rapidly released and then completely released within 7 h. These results clearly show the difference in the release pattern of 5-FU according to the pH.

This pattern occurs because of the carboxylic acid groups present in SG and CMC. At pH 1.2, the hydrogel shrinks because the carboxyl group is protonated, and hydrogen bonding is strengthened. Therefore, it can slow the release of the drug. On the other hand, at pH 7.4, the release rate is increased by increasing the hydrophilicity of the polymer because the carboxylic acid group is ionized. Therefore, these results clearly suggest that SG/CMC IPN gels with higher drug release under neutral conditions can be used to deliver drugs to carriers [53]. This is the first report of pH-dependent drug release in gel form instead of bead form using unmodified natural polysaccharide. Appendix A summarizes the results of other previously published papers.

### 3.8. Cytotoxicity of Hydrogels

The manufactured hydrogel must be non-toxic for use in drug delivery and biomedical applications. The in vitro cytotoxicity of SG/CMC IPN gels was assessed using a WST-8 assay. Cell viability was evaluated and investigated by adding 5 mg of hydrogel to the cultured HEK-293 cells, followed by further culturing [54]. As shown in Figure 10, the cell viability of HEK-293 cells after 2 days of additional culture was more than 99.5% for all the hydrogels, showing no significant difference from the negative control group. In particular, in the case of the three SG/CMC IPN gels, cell viability was similar to that of CMC gels, which is known to be non-toxic. It has been reported that hydrogels that form IPN through Fe^3+^ coordination do not have cytotoxcity [55,56,57,58,59]. On the other hand, the positive control treated with DMSO showed a cell viability of 25%. The results indicate that the SG/CMC IPN gel does not cause negative effects on HEK-293 cells and can be used for biomedical applications such as drug delivery.

## 4. Conclusions

In this study, SG/CMC hydrogels were prepared by applying the IPN polymerization of SG to CMC gels via a double cross-linking strategy of two polymers using ionic cross-linking between carboxyl groups and Fe^3+^. This hydrogel enables an effective pH-responsive drug delivery with significantly improved mechanical strength while maintaining the super absorbency, which is the unique advantage of CMC gel. The resulting SG/CMC IPN gel could control the rheological properties, the cross section, and pore size of the gels, depending on the relative ratio of SG. In addition, the prepared hydrogel exhibited a pH-responsive swelling property, and the drug release of 5-FU where the release was significantly increased at a physiological pH (pH 7.4) compared to an acidic condition (pH 1.2). In addition, the SG/CMC IPN hydrogel was not cytotoxic. Overall, our results mentioned above suggest that the prepared SG/CMC IPN hydrogels have potential for use in a variety of biomedical applications such as cosmetics, food engineering, and controlled drug delivery systems.

## Figures and Tables

**Figure 1 polymers-13-03197-f001:**
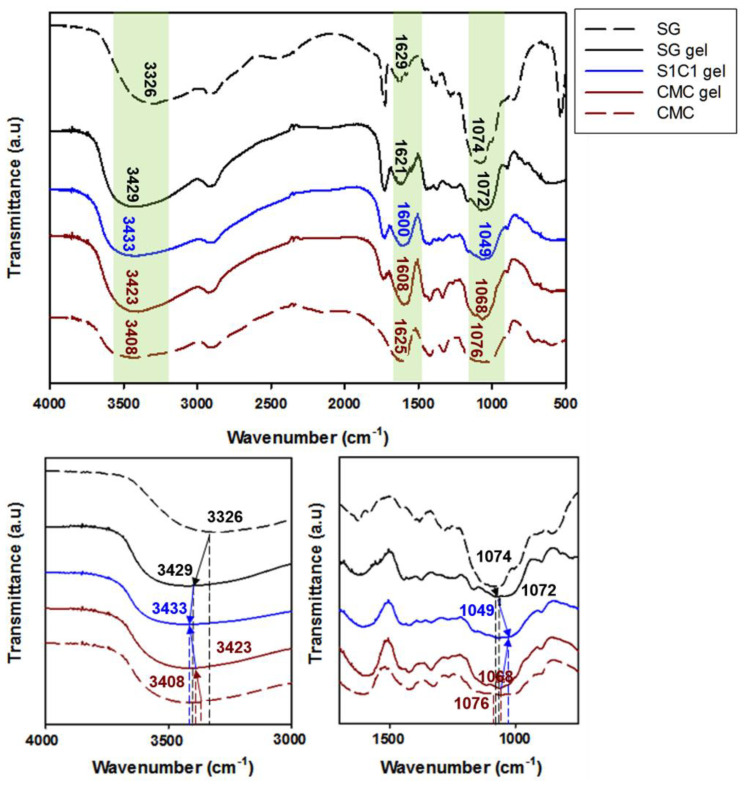
FTIR spectra of SG gel, S1C1 gel, and CMC gel.

**Figure 2 polymers-13-03197-f002:**
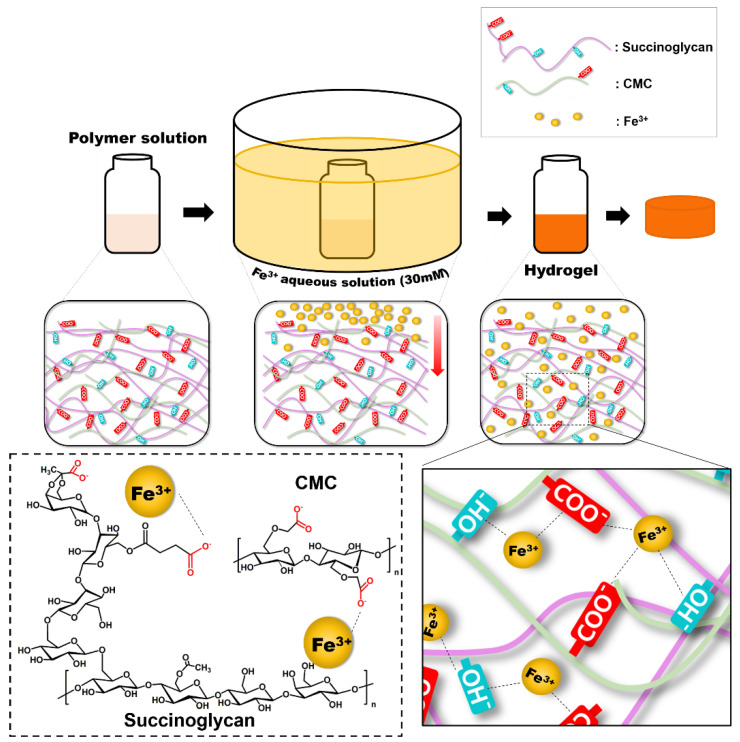
Schematic illustration of model of SG/CMC IPN hydrogels formation mechanism.

**Figure 3 polymers-13-03197-f003:**
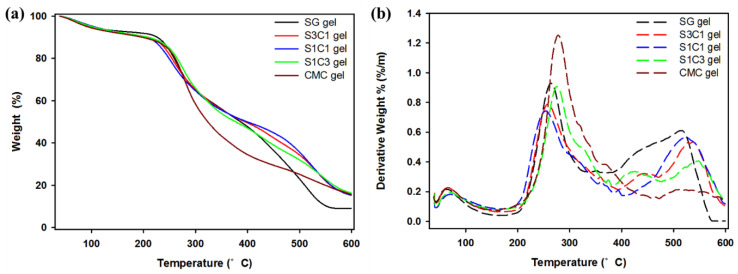
(**a**) TGA and (**b**) DTG curves of SG/CMC IPN gel.

**Figure 4 polymers-13-03197-f004:**
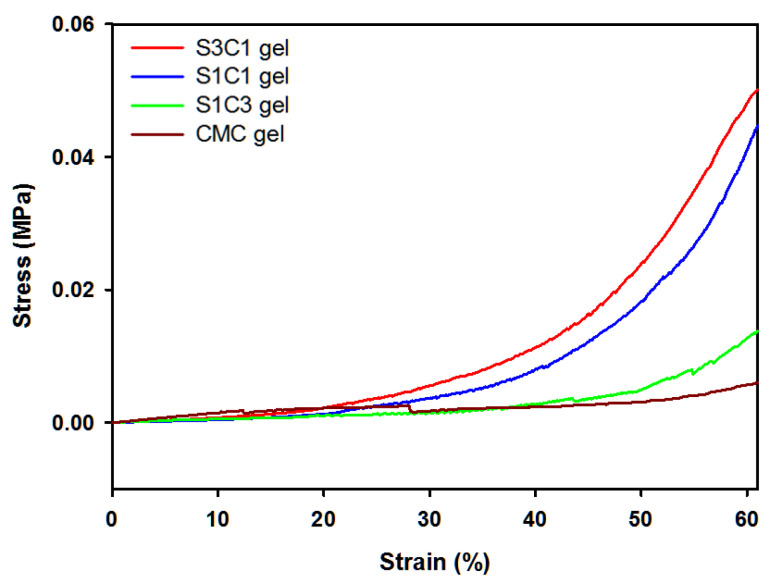
Compressive stress–strain curves of hydrogels with different SG concentrations.

**Figure 5 polymers-13-03197-f005:**
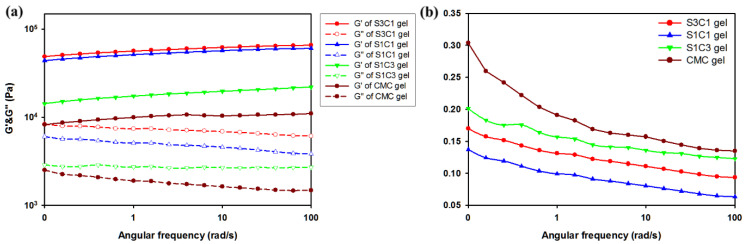
Rheological observations from the oscillation angular frequency sweep test: (**a**) Storage modulus (G′, filled symbols) and loss modulus (G″, empty symbols) of hydrogels; (**b**) The loss tangent (tan δ) of hydrogels.

**Figure 6 polymers-13-03197-f006:**
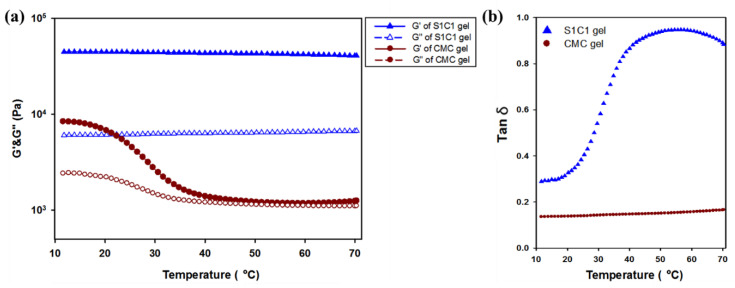
Temperature dependence of hydrogel through rheological observation of the oscillation temperature ramp test. (**a**) Storage modulus (G′, filled symbols) and loss modulus (G″, empty symbols) of hydrogels. (**b**) The loss tangent (tan δ) of hydrogels.

**Figure 7 polymers-13-03197-f007:**
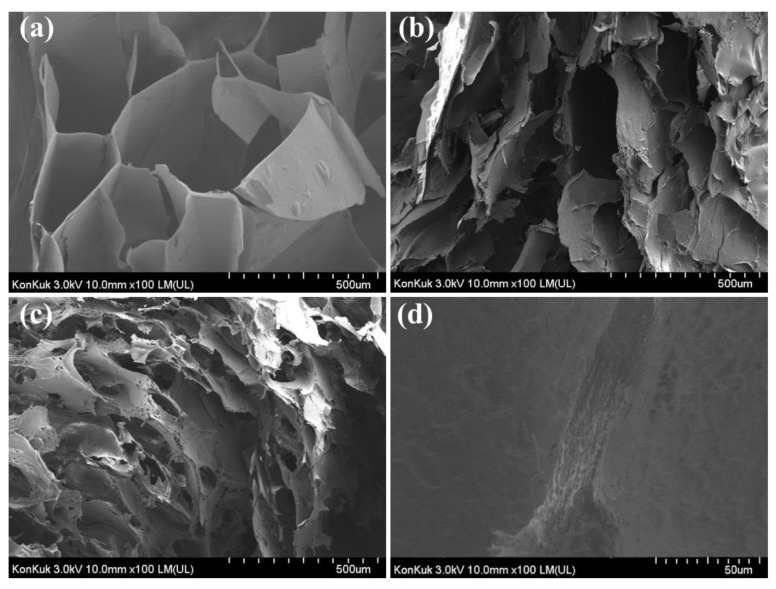
FE-SEM images illustrating the microstructures of cross-sectioned hydrogels: (**a**) SG gel; (**b**) S1C1 gel (Low magnification); (**c**) CMC gel; (**d**) S1C1 gel (High magnification).

**Figure 8 polymers-13-03197-f008:**
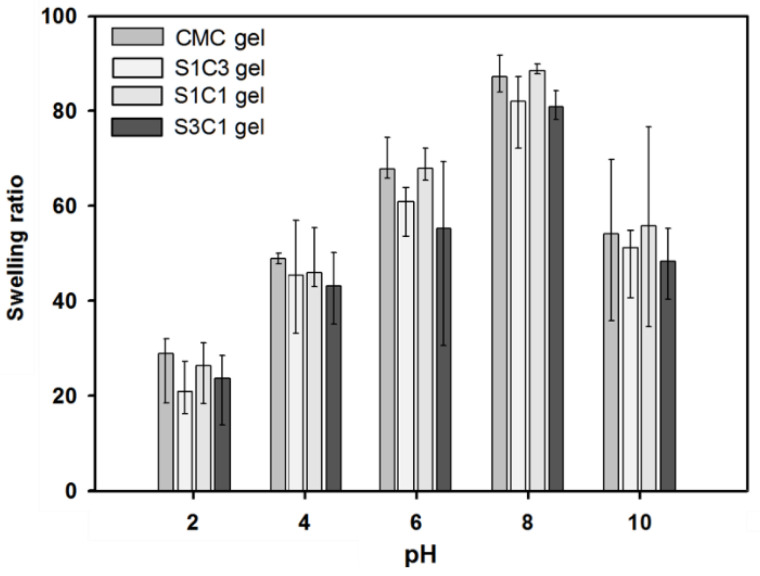
The influence of different pH conditions (pH 2, 4, 6, 8, and 10) on the equilibrium swelling ratio of SG, CMC gel, and SG/CMC IPN hydrogel at 37 °C.

**Figure 9 polymers-13-03197-f009:**
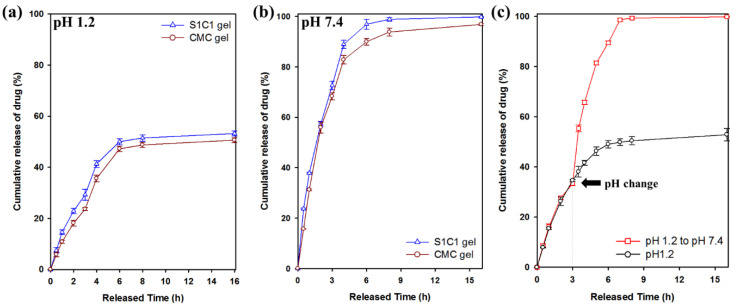
Cumulative release of 5-FU from of SG, CMC gel, and SG/CMC IPN hydrogels at 37 °C: (**a**) pH 1.2 condition; (**b**) pH 7.4 condition; (**c**) Change the pH from 1.2 to 7.4 after 3 h of experiment.

**Figure 10 polymers-13-03197-f010:**
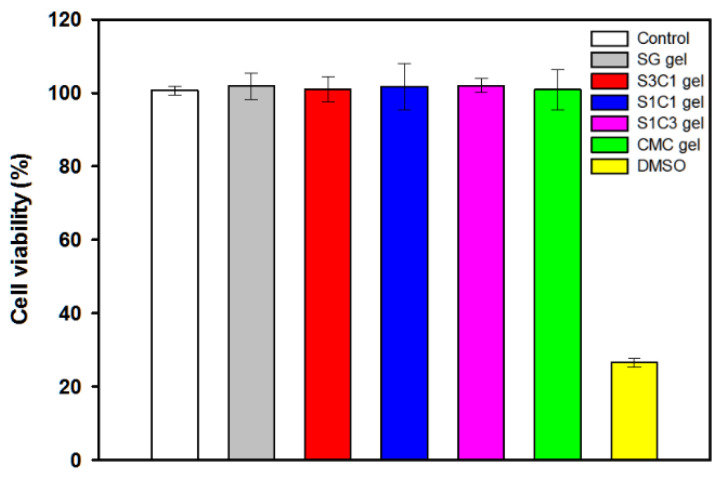
Cytotoxicity of SG/CMC IPN hydrogels against HEK 293 cells.

**Table 1 polymers-13-03197-t001:** Composition of Composition of SG/CMC IPN gel.

Sample Name	Succinoglycan (%)	CMC (%)	Concentration of FeCl_3_ (mM)
**SG gel**	100	-	30
**S3C1 gel**	75	25	30
**S1C1 gel**	50	50	30
**S1C3 gel**	25	75	30
**CMC gel**	-	100	30

**Table 2 polymers-13-03197-t002:** Shift of the main peaks in the FTIR spectra of SG gel, S1C1 gel, and CMC gel.

Peak Assignment		Peak Wavenumber (cm^−1^)			Reference
		SG	CMC	S1C1	
OH stretch		3326	3408	3433	[36,37,39,41]
C=O stretch	Acetate	1728	-	1600	[36,37,39,40]
	Succinate/pyruvate	1629	-		
	Carboxymethyl ether	-	1625		[40]
asymmetric C-O-C stretch		1074	1076	1049	[36,37,44]

## Data Availability

Not applicable.

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
