# Peer review of "pH-Responsive Succinoglycan-Carboxymethyl Cellulose Hydrogels with Highly Improved Mechanical Strength for Controlled Drug Delivery Systems"

_polymers, 2021, doi:10.3390/polym13183197_

Round 1

Reviewer 1 Report

  1. The paper is not of high quality, and major revision is requested before possible publication. The abstract could be shortened and rewritten to show the originality and or the interesting results.
  2. The Introduction section needs more detail about the subject. The authors should review the other new investigation on their study way in the introduction part and finally note the novelty of the article. The authors should note directly to the main goal of the research in a sentence at the end of the introduction.
  3. Results and discussion: It is one of the weakest parts of the manuscript. Also, authors cloud compares the results with other published work and I Suggest that authors add a table summarizing the results obtained in this work with other previously published papers.
  1. The mechanism of SG CMC polymer solutions needs to be proof. I think this manuscript needs to more tests and analyses. in this style, the quality of the manuscript is poor. E.g. Compression mechanical properties of SG/CMC IPN hydrogels and Cytotoxicity of hydrogels sections need to more details.
  2. The language used in the manuscript can be more specific to the scope and aim of the study.

Reviewer 2 Report

The paper pH-responsive succinoglycan-carboxymethyl cellulose hydrogels with highly improved mechanical strength for controlled drug delivery systems, prepared by Younghyun Shin, present novel and interesting results that deserve to be published after some minor improvements:

  1. please increase the font size of the text related to each figure.
  2. please format the references according to the journal style.
  3. please add more 5 references from 2019-2020 in order to highlight better the importance of your work in the field.

after that, the paper can be published

Round 2

Reviewer 1 Report

The paper has been improved and corresponding modifications have been conducted. I think, the current version can be considered for publication.